# Genotypic Diversity, Antibiotic Resistance, and Virulence Phenotypes of *Stenotrophomonas maltophilia* Clinical Isolates from a Thai University Hospital Setting

**DOI:** 10.3390/antibiotics12020410

**Published:** 2023-02-18

**Authors:** Orathai Yinsai, Manu Deeudom, Kwanjit Duangsonk

**Affiliations:** Department of Microbiology, Faculty of Medicine, Chiang Mai University, Chiang Mai 50200, Thailand

**Keywords:** *Stenotrophomonas maltophilia*, mutilocus sequence typing, antibiotic resistance, multidrug resistance, biofilm, motility, toxin, enzyme

## Abstract

*Stenotrophomonas maltophilia* is a multidrug-resistant organism that is emerging as an important opportunistic pathogen. Despite this, information on the epidemiology and characteristics of this bacterium, especially in Thailand, is rarely found. This study aimed to determine the demographic, genotypic, and phenotypic characteristics of *S. maltophilia* isolates from Maharaj Nakorn Chiang Mai Hospital, Thailand. A total of 200 *S. maltophilia* isolates were collected from four types of clinical specimens from 2015 to 2016 and most of the isolates were from sputum. In terms of clinical characteristics, male and aged patients were more susceptible to an *S. maltophilia* infection. The majority of included patients had underlying diseases and were hospitalized with associated invasive procedures. The antimicrobial resistance profiles of *S. maltophilia* isolates showed the highest frequency of resistance to ceftazidime and the lower frequency of resistance to chloramphenicol, levofloxacin, trimethoprim/sulfamethoxazole (TMP/SMX), and no resistance to minocycline. The predominant antibiotic resistance genes among the 200 isolates were the *smeF* gene (91.5%), followed by *bla*_L1_ and *bla*_L2_ genes (43% and 10%), respectively. Other antibiotic resistance genes detected were *floR* (8.5%), *intI1* (7%), *sul1* (6%), *mfsA* (4%) and *sul2* (2%). Most *S. maltophilia* isolates could produce biofilm and could swim in a semisolid medium, however, none of the isolates could swarm. All isolates were positive for hemolysin production, whereas 91.5% and 22.5% of isolates could release protease and lipase enzymes, respectively. In MLST analysis, a high degree of genetic diversity was observed among the 200 *S. maltophilia* isolates. One hundred and forty-one sequence types (STs), including 130 novel STs, were identified and categorized into six different clonal complex groups. The differences in drug resistance patterns and genetic profiles exhibited various phenotypes of biofilm formation, motility, toxin, and enzymes production which support this bacterium in its virulence and pathogenicity. This study reviewed the characteristics of genotypes and phenotypes of *S. maltophilia* from Thailand which is necessary for the control and prevention of *S. maltophilia* local spreading.

## 1. Introduction

*Stenotrophomonas maltophilia* is a Gram-negative obligate aerobic bacillus that is often recovered from various environments such as soil, plant, water, and drinking water [1,2,3]. It is widely recognized as an opportunistic pathogen that can cause severe infections in hospitalized patients, especially those with severely impaired immune systems, including those with HIV infection and malignancy, chemotherapy-treated patients, and patients who used immune suppressive drugs [4,5,6]. *S. maltophilia* infections have been increasingly reported worldwide [1,6,7]. In several countries, most cases are nosocomial infections, that are caused by the contamination of medical equipment and water systems in hospitals [8,9,10].

*S. maltophilia* produces many virulence factors that contribute to infection [11]. It usually forms biofilm as a means of attaching to the surface of medical equipment, like respirators and catheters, which promotes their survivability and pathogenicity [1,11]. Another factor is flagella, which are used for motility, and can stimulate the host’s immune response [12]. This bacterium has extracellular enzymes that are released to promote pathogenesis, such as proteases, lipases, esterase, DNase, Rnase, and fibrolysin [6,13]. Due to the exceptional innate antimicrobial resistance of the species and acquired resistance to numerous antimicrobial drugs, treating *S. maltophilia*, infections can be challenging [14]. The multidrug resistance mechanisms of *S. maltophilia* are the production of drug-degrading enzymes and efflux pumps, as well as the transport proteins involved in the extrusion of drugs [15]. TMP/SMX is the first-line treatment for *S. maltophilia* infections, however, substantial rates of TMP/SMX resistance have been increasingly reported [7]. Levofloxacin and minocycline are additional antibiotics against *S. maltophilia* [16].

Recent investigations have revealed the high genetic diversity among *S. maltophilia* strains isolated in different parts of the world [17]. Molecular methods are used to provide evidence of epidemiological relationships between isolates. These methods are also an important tool in the investigation of the spread of *S. maltophilia* infections all over the world. There are a number of studies about *S. maltophilia* genotypes in many countries, which help in the understanding of the epidemiology and clonality of bacteria. Nowadays, many methods have been developed to clarify bacterial genetic background. The gold standard technique is multilocus sequence typing (MLST). MLST is a procedure for the characterization of bacterial species, using seven housekeeping genes’ internal fragment sequences. MLST is used to provide a portable, accurate, and highly discriminating typing system that can be used for most bacteria and some other organisms [18]. The unique allele sequences of the seven housekeeping genes were defined as allelic profiles or STs [19]. Phenotypic characterization of *S. maltophilia* includes growth rate, biofilm formation, motility, mutation frequencies, antibiotic resistance, virulence, and pathogenicity. These phenotypic factors are required to give more understanding of the relationship between genotypic patterns and phenotypes within the bacterial population.

*S. maltophilia* from different countries or regions may have different genotypic properties with the divergence of virulence genes, drug resistance genes, and mobile genetic elements [20,21,22,23]. These genotypic differences may contribute to the differences in phenotypic properties, e.g., morphology, bacterial pathogenesis and virulence, drug resistance property, gene exchange or gene transfer, enzyme production, and biofilm formation. During the last decade, *S. maltophilia* has been recovered with increasing frequency at Maharaj Nakorn Chiang Mai Hospital, a 1400-bed university-affiliated hospital in Chiang Mai, Thailand [24]. Moreover, published data is rarely available on the epidemiology and characteristics of *S. maltophilia* in Thailand [25].

Therefore, this study aims to characterize clinical information, genotypes, and phenotypes of *S. maltophilia* isolated from Maharaj Nakorn Chiang Mai Hospital. The investigation includes molecular classification based on housekeeping gene allelic patterns using MLST, and the identification of antibiotic resistance genes such as β-lactamase encoding genes, efflux pump encoding genes, and the integrase gene. In addition, the phenotypic properties of *S. maltophilia,* including antibiotic resistance patterns, biofilm formation, motility, and enzyme production were investigated.

## 2. Results

### 2.1. Bacterial Collection and Clinical Characteristics

A total of 200 *S. maltophilia* nonrepetitive isolates were randomly collected from Maharaj Nakorn Chiang Mai Hospital, Chiang Mai, Thailand for six months. In our collection, 199 isolates were collected from patients and one isolate, from a hospital environment (fluid from the dialysis unit). All clinical isolates were obtained from four different sources and most of the isolates were from sputum (*n* = 152; 76%), while others were from body fluids (*n* = 21; 10.5%), pus (*n* = 20; 10.0%), and urine (*n* = 6; 3.0%).

Among *S. maltophilia*-infected patients, 192 patient information could be accessed (Table 1). In total, 117 male patients (60.94%) and 75 female patients (39.06%) were included. The age of the youngest patient was one month, and the oldest patient was 99 years. The average of the ages was 54.61 ± 28.14 years, and the aged people (≥65 years) were the most affected group (*n* = 79; 41.15%).

All patients had at least one underlying illness and were exposed to predisposing factors such as invasive procedures, surgery, chemotherapy, and radiotherapy. The most common underlying diseases and comorbidities were malignancy (*n* = 56; 29.17%), CNS and cerebrovascular diseases (*n* = 36; 18.75%), and urinary tract infections (*n* = 28; 17.71%). One hundred and ninety patients were hospitalized (98.96%) in different wards, including general medicine (*n* = 53; 27.60%), pediatric (*n* = 33; 17.19%), general surgery (*n* = 24; 12.5%), emergency surgery (*n* = 17; 8.85%), orthopedics (*n* = 9; 4.69%), neurosurgery (*n* = 3; 1.56%), and other wards (*n* = 51; 26.56%).

Since the majority of the isolates were obtained from respiratory specimens, risk factors associated with respiratory tract infection were evaluated. The patients who were significantly more likely to develop respiratory tract infections included aged patients, patients with malignancy, CNS and cerebrovascular diseases, and cardiovascular diseases (Table 1). Furthermore, the respiratory group had a significantly higher number of patients who were exposed to intravenous catheters, surgery, chemotherapy, and radiotherapy, and were admitted to the intensive care unit.

### 2.2. Antibiotic Susceptibility and Antibiotic Resistances

The MIC results show that *S. maltophilia* was highly resistant to CAZ (*n* = 155; 77.5%). All the isolates that showed a lower resistance frequency were observed to C (*n* = 36; 18%), LEV (*n* = 18; 9%), TMP/SMX (*n* = 15; 7.5%), and no resistance to minocycline. The MIC range, MIC_50_, and MIC_90_ values are shown in Table 2. From all of the isolates, 37 isolates were susceptible to all antibiotics tested. On the contrary, 20 isolates were resistant to three or more antibiotics, hence exhibiting a multidrug-resistant (MDR) phenotype.

Antibiotic susceptibility results revealed six antibiotypes: TMP/SMX resistance, LEV resistance, CAZ resistance, C resistance, nonresistance, and MDR. Most isolates in the TMP/SMX and LEV resistances also exhibited MDR phenotypes at a high rate (86.67% and 72.22% of isolates tested, respectively). MDR phenotypes were found in a lower percentage of C resistance (64.52%). On the contrary, the majority of CAZ-resistant isolates exhibited a non-MDR phenotype at a higher rate than MDR (Appendix A). However, non-MDR negatively correlated with all of the antibiotic-resistant groups, whereas the MDR phenotype showed a positively correlated with three antibiotic-resistant groups, except CAZ resistance.

Antibiotic resistance phenotypes in *S. maltophilia* varied between specimens. The majority of sputum isolates (88.08%) were resistant to CAZ and have a lower proportion of the isolates that are resistant to C (21.85%), TMP/SMX (6.63%), and LEV (7.28%). Sixteen out of twenty isolates from pus were resistant to CAZ (80.0%), two isolates were resistant to C (20.0%), and none were resistant to TMP/SMX and LEV. All isolates from body fluid were resistant to CAZ and four isolates were resistant to C and TMP/SMX. However, resistance to LEV, was not found. All of the isolates from urine were resistant to CAZ and four out of six isolates were resistant to C, TMP/SMX, and LEV. Minocycline was the most active compound tested against our isolates with no resistance among all isolates or specimens (MIC_50_ of 0.5 µg/mL and MIC_90_ of 2.0 µg/mL). Interestingly, only one isolate from the hospital environment did not show resistance properties to any of the drugs tested.

### 2.3. Detection of Antibiotic Resistance Genes

The detection of antibiotic resistance genes is shown in Table 3. The *smeF* gene was found in the majority of the isolates (91.5%), while *sul2*, *floR*, and *mfsA* were only found in a few (2%, 4%, and 4.5%, respectively). The *bla*_L1_ and *sul1* genes were found in greater proportion (43% and 6%) among resistant groups than the *bla*_L2_ and *sul2* genes (10% and 2%). The *intI1* gene, on the other hand, was found in all resistant groups. The majority of the *sul1*-*2* and *intI1* positive isolates were also TMP/SMX resistant.

### 2.4. Biofilm Formation

The results from the biofilm formation assay showed that most of the isolates are strong biofilm producers (*n* = 146; 73%), whereas one strain (0.5%) did not form biofilm. Moderate biofilm was formed by 31 isolates (15.5%), while 22 strains were weak biofilm producers (11.0%) (Table 4). The comparison of biofilm formation efficiency among the six drug-resistant groups revealed that isolates in the nonresistant group produced significantly more biofilm than isolates in the TMP/SMX, C, and MDR groups (Figure 1A). In all drug resistance patterns, the majority of isolates produced strong biofilm levels, followed by moderate and weak levels, respectively. The biofilm formation capability of isolates, compared among four types of specimens, was not significantly different (Figure 1B).

### 2.5. Motility

Swimming motility was observed in most of the isolates. Twenty-one (10.5%) isolates were nonmotile, while 73 (36.5%) exhibited weak motility, and 86 (43.0%) moderate motility (Table 4). Swimming motility was compared in the six drug resistance patterns. Swimming efficiency was significantly higher in nonresistant and CAZ-resistant isolates than in TMP/SMX, C, and MDR isolates (Figure 2A). In all antibiotic resistance patterns, the majority of isolates could swim moderately or weakly, followed by no swimming and strongly swim, respectively. When the isolates from different types of specimens were compared, it was discovered that the swimming ability of the isolates from pus (high viscosity environment) was significantly higher than the isolates from other specimens (Figure 2B). However, swarming motility was not detected in any of the isolates.

### 2.6. Toxin and Enzyme Production

Screening of toxin and enzyme production revealed that all isolates could produce hemolysin after an incubation time of 48 h on a 5% sheep’s blood agar plate. In 53% of the isolates (*n* = 106), a greenish zone appeared around the bacterial colony, indicating that those isolates could produce α-hemolysin, while 94 isolates (47%) could produce β-hemolysin. Protease enzyme was found in 183 of 200 isolates (91.5%), whereas lipase production was observed in 22.5% of isolates (*n* = 45) after 48 h. Among the various drug resistance patterns, isolates in the majority of groups produced more α-hemolysis than β-hemolysis. Protease-positive isolates were common in all antibiotic resistance groups, while a few isolates produced lipase enzymes (Table 4). There were 21 isolates capable of producing all three enzymes. These isolates were obtained from sputum (17 isolates), fluid (3 isolates), and pus (1 isolate), in that order.

### 2.7. Correlation of Antibiotic Resistance

The relationship between antibiotic resistance genotypes and phenotypes showed that the presence of *sul1*, *sul2*, *intI1*, and *floR* genes (Figure 3) were positively correlated with each other as shown in high Spearman *r* values. Among antibiotic-resistant groups, C resistance was correlated with TMP/SMX resistance (Spearman *r* = 0.50, *p* < 0.0001) and LEV resistance (Spearman *r* = 0.46, *p* < 0.0001), but not with CAZ resistance (Spearman *r* = 0.01, *p* = 0.0001). Furthermore, a positive correlation was discovered between the presence of drug-resistance genes and antibiotic-resistance properties. TMP/SMX resistance was linked to the presence of *sul1*, *sul2*, *intI1*, and *floR*. Chloramphenicol resistance was also found to be associated with the *sul1* and *intI1* genes.

### 2.8. MLST Analysis and Clonal Complexes

*S. maltophilia* allelic profiles revealed 141 STs across 200 isolates. The profiles were created using different patterns of allelic numbers at seven different loci. There were 11 STs among 16 isolates; ST3, ST4, ST24, ST27, ST28, ST77, ST91, ST208, ST210, ST212 and ST511 have been reported on database previously. However, 130 STs among 184 isolates were reported for the first time in this study (ST365, ST376, ST605, ST609, ST611, ST613, ST615, ST618-619, ST621, ST626-628, ST631-632, ST634, ST639, ST643-648, ST651, ST656-660, ST663-669, ST671-675, ST678-681, ST684-688, ST692, ST697-698, ST700, ST703-705, ST709, ST713-714, ST718, ST720-721, ST731, ST736-738, ST745, ST748-750, ST752-754, ST756-764, ST766-770, ST773, ST775, ST777, ST780-785, ST788-808, and ST810-818). These 130 new STs and allelic profiles from Thailand have been submitted to PubMLST (https://pubmlst.org/organisms/stenotrophomonas-maltophilia accessed on 7 December 2022). The most common ST was ST619, which was found in six isolates, followed by ST672 and ST749 in five isolates, respectively. Additionally, the ST type of the isolates from the hospital environment (ST793) was similar to the isolate from human fluid in this collection.

Clonal complexes were studied using goeBURST analysis. *S. maltophilia* isolates from this study were classified into six clonal groups, based on the variation of allelic profiles (Figure 4) The six clonal groups exhibited different genotypes and phenotypes. Characteristics of the majority of isolates in each group are shown in Table 5. The population distribution of *S. maltophilia* worldwide was also analyzed as a minimum-spanning tree, using all *S. maltophilia* in the MLST database. *S. maltophilia* from Thailand was distributed in many branches of the tree as shown in Figure 5. *S. maltophilia* isolates from Thailand (e.g., ST646, ST709, ST818) were found to be closely related to *S. maltophilia* from Asian countries such as China, Korea, and Japan (classified in the same branch). Similarly, some of the isolates (e.g., ST365, ST635, ST700, and ST734) are related to those from Europe and America such as isolates from the UK, France, Mexico, and the USA.

## 3. Discussion

*S. maltophilia* infection has become an important, emerging opportunistic pathogen, which has been increasingly reported worldwide [13]. Thailand also has reported a large number of *S. maltophilia* infections, but the characterization of this pathogen of concern is rarely found [24].

In this study, 200 *S. maltophilia* isolates were collected from Maharaj Nakorn Chiang Mai Hospital, Thailand during six months of collection. Most of the isolates were collected from sputum. The collection of isolates in a short period of time indicates that the incidence of *S. maltophilia* respiratory tract infections in Thailand is high compared to other regions of the world [25,27,28,29,30,31,32].

The clinical information showed that male and elderly patients were more susceptible to *S. maltophilia* infection, frequently distributed in the general medicine ward. The majority of isolates were from hospitalized patients who suffered from underlying illnesses associated with invasive procedures. The patients with respiratory tract infections exhibited a higher proportion of risk factors than non-respiratory tract infection patients. The retrospective studies from China and the USA also reported demographic and clinical characteristics of *S. maltophilia* infection, similar to this study [33,34]. These findings should serve as a reminder to clinicians to focus more on *S. maltophilia* infection control in specific population groups [33]. However, the isolates from this study were collected at the high prevalence period, however, the obtained data were not up to date. The efficient strategy of infection control should be carried out together with the information from the recent isolates in further study.

Antibiotic susceptibility revealed that the high resistance incidence to CAZ of Thai *S. maltophilia* was distinguished, compared to the global trend [35]. Although TMP/SMX is the current drug of choice for *S. maltophilia* treatment with a high sensitivity (79–96%), the resistance has been raised worldwide (30–48%) [16,36,37,38,39]. The isolates of our study showed a similar rate of TMP/SMX resistance to those of global isolates [34]. Minocycline is the most active antibiotic against the isolates from our study and another geological region of Thailand and China [33,40].

It must be noted that our *S. maltophilia* isolates from urinary tract infections are highly resistant to antibiotics compared to isolates from other specimen sources, which is similar to a study by Hamdi et al. from Minnesota, USA [41]. According to antibiotic susceptibility patterns, there were six antibiotypes in which isolates with TMP/SMX and LEV resistance appeared to have MDR phenotypes in higher proportion than isolates with C and CAZ resistances. This result was consistent with the study of Zhao et al., which found that TMP/SMX resistance was a signal of multidrug resistance [42].

Numerous molecular processes contribute to *S. maltophilia*’s widespread antibiotic resistance. The *smeDEF* genes are an efflux pump encoding protein complex of *S. maltophilia*, that are involved in quinolones, chloramphenicol, and tetracyclines resistances [5]. In our findings, *smeF* was the most common gene (91.5%) among our collection of isolates, of which 89% of them were LEV and C resistance. The *bla*_L1_ gene, a Zn^2+^-dependent metalloenzyme that can hydrolyze β-lactams [43], was found in 43% of isolates, and most of them (78.34%) were resistant to CAZ, suggesting that the role of the *bla*_L1_ gene is contributing to β-lactam resistance of this *S. maltophilia* collection. Meanwhile, *bla*_L2_ (a serine active-site cephalosporinase [43]) showed a less important role in CAZ resistance similar to other collections of southern Thailand and Iran [25,44]. Our findings showed that *sul1* and *intI1* were detected in six percent and seven percent of isolates, respectively. All of the *sul1* and *intI1* positive isolates were resistant to TMP/SMX, indicating that TMP/SMX resistance was mediated by *sul1* and class one integron integrase genes [45,46]. Bostanghadiri et al. similarly found a higher rate of *sul1*-positive strains among isolates from Iran [44]. In addition, the Florfenicol/chloramphenicol resistance gene, *floR*, and a major facilitator superfamily (MFS) of the efflux pump gene, *mfsA* could also be detected in eight and eight point five percent of *S. maltophilia* collection. The correlation analysis revealed that *sul1*, *sul2*, *IntI1*, and *floR* genes are positively correlated to each other and involved with TMP/SMX resistant phenotypes, similar to a prior study from Nigeria [47].

Among *S. maltophilia* virulence factors, biofilm plays an important role in the survivability and virulence of many bacteria, as is found in *S. maltophilia*. In this study, all *S. maltophilia* isolates were able to produce biofilm and most of them were strong biofilm producers (73.0%). Interestingly, the nonresistance showed significantly higher level of biofilm formation compared to the other groups. There were similar findings in 2020, which found that non-MDR *S. maltophilia* exhibited higher biofilm formation capacity compared to MDR phenotypes [48]. This suggested that biofilm-forming *S. maltophilia* isolates depended less frequently on antibiotic resistance for survival as those isolates do not need the biologically costly expression of antibiotic resistance to survive in an environment, such as a hospital setting [48].

In the study of motility, most of the isolates exhibited weak or moderate levels of swimming phenotypes, however, none of the isolates were able to swarm, which is similar to some other studies [49,50]. Surprisingly, *S. maltophilia* isolates from urine exhibited significantly less swimming capability than isolates from other sources. This incidence may be affected by the different densities of biological environments, in which, the bacterium swims faster in a suspension with more viscosity (e.g., sputum, pus, and body fluid) [51]. Moreover, the transitions between motility and adherent state were found in UTI-causing bacteria which helped the improved colonization of bacteria to the upper urinary tract [52]. In our study, all *S. maltophilia* isolates could produce hemolysin, and α-hemolysin was frequently detected, while most of the isolates in our collection were able to produce protease (91.5%), whereas the number of lipase-producing isolates were found in a lower proportion of isolates (22.5%).

The study of the genetic relationship and MLST analysis in *S. maltophilia* isolates from Chiang Mai, Thailand showed high diversity in allelic profiles. The majority of isolates contained new allelic sequences that have never been reported before. We submitted to an international database and reported 130 novel STs. This study also found 11 STs that were previously reported in the studies from Japan, Korea, and Germany [5,19,53], especially ST77, which was distributed widely throughout the world. The MLST profiles of *S. maltophilia* from several countries even in the recent study from Iran were similar to this study in terms of the great diversity of STs in a single hospital [44,54,55].

Moreover, our study included one isolate from a hospital environment, and this isolate belongs to ST739, which is the same ST as the one isolate from a human patient. This incidence supported the study of Gideskog et al. which found the clonal relationship between isolates from patients and hospital settings, which help them achieve infection control by replacing contaminated devices [56]. The *S. maltophilia* outbreak most likely depended on the environmental spread and further study of the genetic relationship between isolates from specimens and hospital environments is required to promote an understanding of the *S. maltophilia* hospital outbreak and control the infection.

From genetic population analysis of global *S. maltophilia*, seven major clonal complexes exhibited that *S. maltophilia* from Thailand was closely related to *S. maltophilia* from other countries such as China, South Korea, Japan, and the USA. Interestingly, Thai isolates were dominant and found as founders in some branches of clonal complexes which were considered to be ancestors and a reservoir of *S. maltophilia*. However, this assumption needs to be confirmed with genetic population data of a larger number of *S. maltophilia* from Thailand. The clonal complex analysis of Thai isolates identified six major different groups based on different allelic sequences and multilocus variants. Each group carried different antibiotic resistance genes and exhibited different phenotypes. Therefore, there was no association between genetic lineages and *S. maltophilia* phenotypes. Similarly, the novel STs had been identified from a previous study and they also found differences in resistance and virulence genes in their collection [57]. The association of clonal complex and particular specimens of isolation were not observed, except the isolates from urine which were found in the same clonal complex and correlated with the MDR phenotype.

There are a number of limitations to this study. In order to understand the epidemiology of *S. maltophilia* isolates in hospitals and implement infection control strategies, additional studies that include more recent isolates and environmental sampling are required. Moreover, additional epidemiological multicenter studies in Thailand with extended surveillance are required to better characterize the prevalence and spread of nosocomial infections linked to *S. maltophilia*.

## 4. Materials and Methods

### 4.1. Bacterial Collection, Culture, and Clinical Information

*S. maltophilia* isolates were randomly collected from the Microbiology Unit, Diagnostic Laboratory, Maharaj Nakorn Chiang Mai Hospital, Chiang Mai, Thailand for six months, from October 2015 to March 2016. This collection included clinical isolates which were isolated from various patient specimens and environmental isolate which was collected from a hospital environment. Each isolate was collected per one patient or one environmental sample and must be the dominant bacteria with significant numbers, not a contaminant. All the isolates were identified by a microbiology unit, and diagnostic laboratory using mass spectrometry (MALDI Biotyper^®^, Bruker Corp., Billerica, MA, USA). Bacterial isolates were cultured on Luria Bertani (LB) agar and incubated overnight at 37 °C. Bacterial stocks are preserved at −80 °C in LB broth containing 20% skimmed milk. This study used *Escherichia coli* ATCC 25922, *Pseudomonas aeruginosa* ATCC 27853, and *Proteus mirabilis* FL118 as control strains. Additionally, demographic information and clinical characteristics including age, gender, underlying diseases, comorbidities, predisposing factors, hospitalization, and medical wards of 192 included patients from our collection were recorded.

### 4.2. Species Confirmation by 23S rRNA PCR

Total DNA was extracted from each isolate by using a Thermo Scientific^TM^ GnenJET nucleic acid purification kit (Thermo Fisher Scientific Inc., Waltham, MA, USA) according to the instruction of the manufacturer. All *S. maltophilia* isolates in this study were confirmed by PCR using specific primers to the 23S rRNA encoding gene (Forward primer: 5′ GCTGGATTGGTTCTAGGAAAACGC 3′; Reverse primer: 5′ ACGCAGTCACTCCTTGCG 3′). Twenty microliters of PCR reaction total volume contains 10 µL of 2X master mix solution (iNtRON Biotechnology Inc., Burlington, MA, USA), 5 µM of each primer condition, and 100 ng of DNA template. Amplification was performed as previously described [58].

### 4.3. Antibiotic Susceptibility

Antimicrobial susceptibility testing of *S. maltophilia* was determined by minimal inhibitory concentration (MIC) using the agar dilution method and MIC test strip. Four antibiotics including levofloxacin (LEV), ceftazidime (CAZ), chloramphenicol (C), and minocycline (MN) were tested by agar dilution. *S. maltophilia* colony suspension was prepared and adjusted equivalent to McFarland no. 0.5 before diluting 1:10 and inoculating on antibiotic-contained MHA using a multipoint inoculator. MIC of TMP/SMX was determined by MIC test strip (TMP/SMX = 1/19, 0.002–32 µg/mL) (Liofilchem s.r.l., Abruzzo, Italy). The bacterial suspension was similarly prepared as agar dilution and was swabbed onto an MHA plate (no drug). *E. coli* ATCC 25922 and *P. aeruginosa* ATCC 27853 were used as quality control strains and the results were interpreted according to Clinical and Laboratory Standards Institute (CLSI) guidelines [26]. Isolates resistant to at least three antibiotics were considered MDR [59,60].

### 4.4. Detection of Drug Resistance Genes

PCR assays were used to detect eight antibiotic resistance genes, including *bla*_L1_/*bla*_L2_ genes, *smeF*, *sul1/ sul2*, *IntI1*, *floR*, and *mfsA*. All primers used are listed in Appendix A [25,30,50,61,62,63,64,65]. The PCR reaction and conditions were carried out as previously described [25]. The amplicons were detected and visualized under ultraviolet light using a 1.5% agarose gel stained with Redsafe (iNtRON Biotechnology Inc., MA, USA).

### 4.5. Biofilm Formation Assay

Overnight culture of *S. maltophilia* in a tryptic soy broth (TSB) at 37 °C was diluted at 100-fold dilution and was transferred into a 96-well plate. After washing and removing bacterial planktonic cells by PBS, the plate was heat fixed at 60 °C for 15 min and stained with 0.1% crystal violet for 5 min. After that, the plate was rinsed three times with water and then 30% acetic acid was added to dissolve the dyed pellet. Biofilm formation capability was observed by measuring optical density (OD) at 595 nanometers. Their observed optical density was classified as follows: no biofilm producer (OD ≤ OD negative control (ODc); weak biofilm producer (ODc ≤ OD ≤ 2 × ODc); moderate biofilm producer (2 × ODc ≤ OD ≤ 4 × ODc); and as a strong biofilm producer (OD > 4 × ODc). The negative controls were wells that contained culture medium alone. *P. aeruginosa* ATCC 27853 was used as a positive control for biofilm production [66].

### 4.6. Motility Test

Ten microliters of bacterial overnight culture in TSB were dropped and stabbed into a swimming medium (containing 10 g/L tryptone, 5 g/L NaCl, 3 g/L agar) and dropped on a swarming medium (containing 8 g/L nutrient broth, 5 g/L). As for the interpretation of the swarming test, a positive result was observed by a transparent growth zone appearing around a bacterial colony. The swimming zones on media were measured in millimeters (mm) and classified by the following criteria: no swimming motility (<3 mm); weak swimming motility (3–5 mm); moderate swimming motility (6–8 mm); and strong swimming motility (≥9 mm) [67]. Positive control strains for these swimming and swarming tests were *E. coli* ATCC 25922 and *P. mirabilis* FL118.

### 4.7. Screening of Toxin and Enzymes Production

Production of toxins and enzymes by *S. maltophilia* was determined as previously described [50]. The isolates were streaked on agar media containing substrates for each enzyme. To test protease enzyme production, the bacterium was inoculated on Mueller Hinton (MH) agar containing 3% skimmed milk and incubated at 37 °C. At 24 h of incubation, the clear zone around the colonies was observed from isolates that could produce protease.

*S. maltophilia* isolates were inoculated on Tryptic Soy (TS) agar containing 1% tween 80, the lipase substrate, and incubated at 37 °C to detect lipase activity. At 48 h, the appearance of a turbid halo zone around colonies indicated a positive outcome. The isolate was streaked on 5% sheep blood agar and incubated at 37 °C for 48 h to detect hemolytic activity. Positive results were seen for β-hemolysin producers when a transparent zone appeared around colonies, and for α-hemolysin producers when a greenish zone appeared around colonies [68].

### 4.8. Multi-Locus Sequence Typing (MLST) Analysis

MLST analysis of *S. maltophilia* was performed as previously described by Kaiser et al. [19]. The alleles at each of the seven loci defined the allelic profile or ST. PCR was performed on *S. maltophilia* isolates using specific primers to seven housekeeping genes, namely, *atpD*, *gapA*, *guaA*, *mutM*, *nuoD*, ppsA, and recA. The primer sequences and PCR condition was set as described by Kaiser et al. PCR products were purified using a Thermo Scientific GeneJET PCR Purification Kit (Thermo Fisher Scientific Inc., MA, USA) and were sequenced. The nucleotide sequences of seven housekeeping genes were compared to the reference sequences on the PubMLST database (https://pubmlst.org/smaltophilia/: accessed on 16 October 2017) for identifying the allelic number of each locus and the classifying STs.

### 4.9. Statistical Analysis

Each microbiological assay was performed at least in duplicate and repeated three times. Mean ± SD was used for the continuous variables. The statistical significance of the difference between groups of the test was calculated by one-way ANOVA and Fisher’s exact test. Continuous variables from demographic and clinical data were analyzed using a Student’s *t* test or the Mann–Whitney *U* test. Correlation analyses were determined by the Spearman test. All statistical tests and graphs were evaluated by GraphPad Prism version 9.1.1. The statistical significance was considered when *p* < 0.05. For genetic population and clonal complex analysis, all given STs were classified into clonal groups by PHYLOViZ software version 2.0 (https://online.phyloviz.net/index: accessed on 23 April 2018), based on the goeBURST algorithm.

## 5. Conclusions

This study revealed demographic, genotypic, and phenotypic characteristics of *S. maltophilia* isolates from a Northern Thailand hospital during the period of the highest prevalence. Infection with *S. maltophilia* can occur in hospitalized patients with a variety of comorbidities and risk factors. We underline a high degree of genetic diversity among the isolates and this is the first report on numerous novel STs of *S. maltophilia* collection in Thailand. Most of the isolates carried many drug-resistance genes and showed a highly resistant rate to several antibiotics, especially, the isolates that were resistant to the drug of choice (TMP/SMX), exhibited MDR phenotype, and also produced various virulence factors. *S. maltophilia* isolates of Thailand were found to be genetically related to *S. maltophilia* from other countries, of which Thai isolates were dominant and found to be a founder. The data obtained from this study contribute to a better understanding of *S. maltophilia* characteristics in Thailand, which is necessary for *S. maltophilia* infection control and prevention.

## Figures and Tables

**Figure 1 antibiotics-12-00410-f001:**
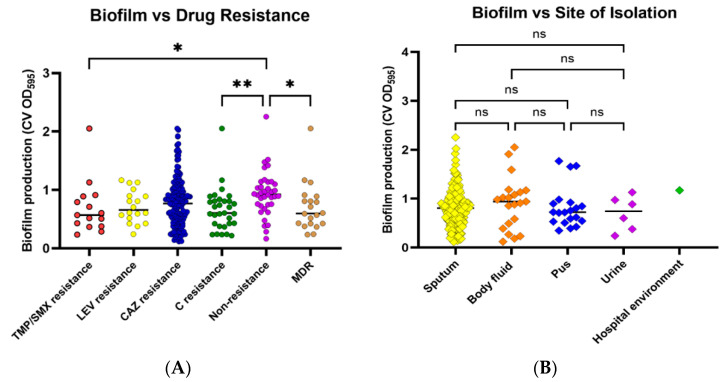
Biofilm formation by *S. maltophilia* isolates. Biofilm formation was evaluated using a crystal violet assay by measuring crystal violet absorbance (CV OD) at 595 nm. Biofilm formation capabilities were investigated among various antibiotic-resistant groups (**A**) and sites of isolation (**B**). Antibiotic-resistant groups included trimethoprim/sulfamethoxazole (TMP/SMX) resistance, levofloxacin (LEV) resistance, ceftazidime (CAZ) resistance, chloramphenicol (C) resistance, nonresistance, and multidrug resistance (MDR). Each symbol showed the mean OD_595_ value with the median line of each distribution. Statistical significance at Fisher’s exact test: * *p* < 0.05, ** *p* < 0.01. ns = non-significant.

**Figure 2 antibiotics-12-00410-f002:**
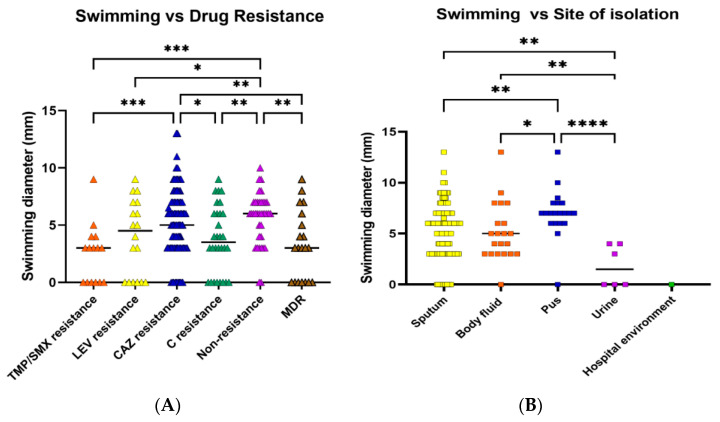
Swimming motility of *S. maltophilia* isolates. Swimming motility was evaluated by measuring the swimming zone diameter in a semi-solid medium (0.3% agar). Swimming efficacies were investigated in various antibiotic-resistant groups (**A**) and sites of isolation (**B**). Each symbol showed the mean swimming diameter with the median line of each distribution. A percentage of isolates belonged to each group. Statistical significance at Fisher’s exact test: * *p* < 0.05, ** *p* < 0.01, *** *p* < 0.001, **** *p* < 0.0001.

**Figure 3 antibiotics-12-00410-f003:**
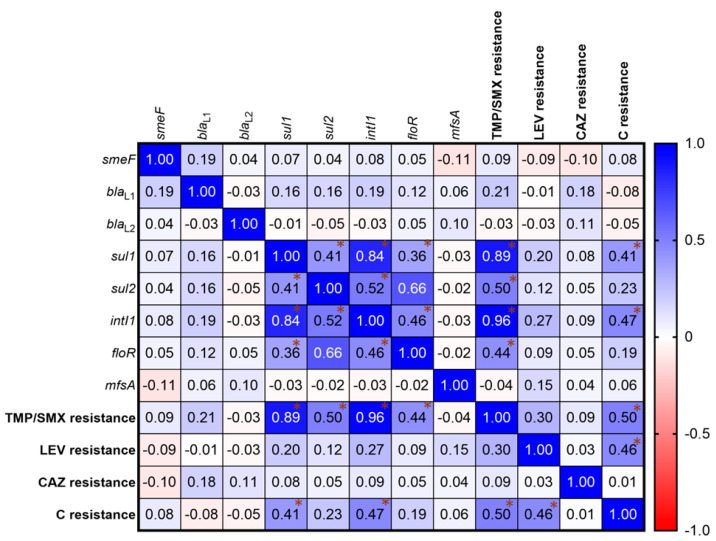
Correlation matrix of *S. maltophilia* antibiotic resistances. The relationships were determined by Spearman correlation coefficients. A heat map shows the Spearman *r* value indicating a correlation between the presence of drug-resistance genes and drug-resistant phenotypes. The gradient of positive and negative values shows a positive correlation (Blue) and a negative correlation (Red). Asterisk (*) shows the *r* value exhibiting the significant correlation. A statistical significance of the correlation was set at *p* < 0.05.

**Figure 4 antibiotics-12-00410-f004:**
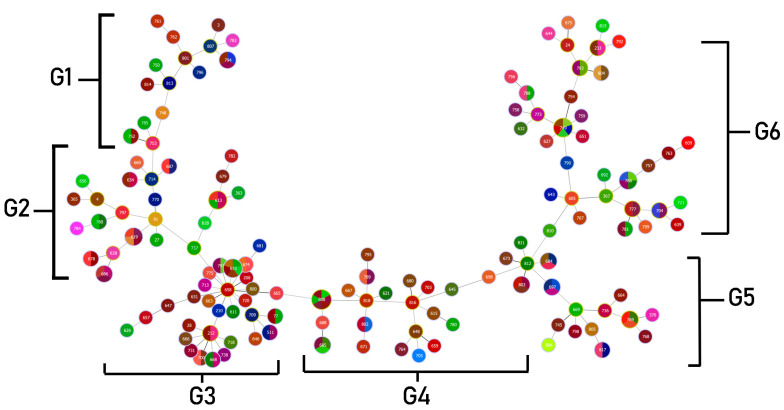
Minimum spanning tree (MST) generated for 200 Thai *S. maltophilia* isolates. The tree was created using PHYLOViZ online software with the goeBURST algorithm. Isolates are represented by different colors and STs are shown in circles. The name of each ST is labeled as the number in the center of the circles. Six clonal complex groups were revealed as G1 (Group 1)–G6 (Group 6).

**Figure 5 antibiotics-12-00410-f005:**
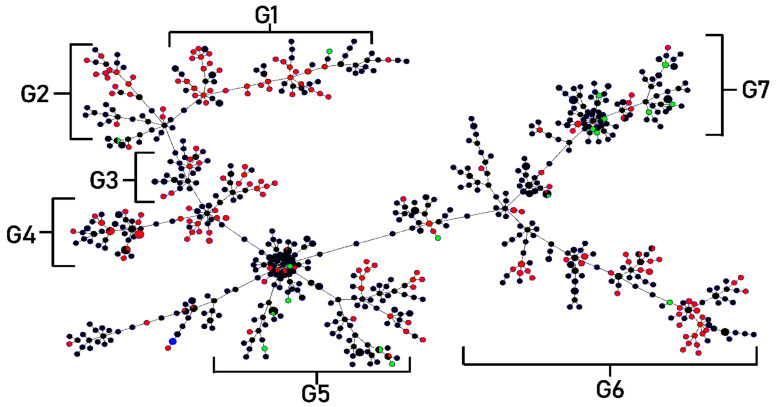
Phylogenetic tree of global *S. maltophilia* isolates based on MLST allelic profile variation. The minimal spanning tree was created using PHYLOViZ online software with the goeBURST algorithm and the analysis considered 995 isolates reported in the *S. maltophilia* MLST database. The population distribution of the isolates worldwide is shown. Countries are shown in different colors and STs in each circle. Isolates from Thailand are represented by pink circles and are distributed all around the branches of the tree.

**Table 1 antibiotics-12-00410-t001:** Demographic and clinical characteristics, and risk factors of respiratory tract infections due to *S. maltophilia*.

Characteristics	No. of Patients (%) (*n* = 192)	No. of Respiratory Infection (*n* = 148)	No. of Non-Respiratory Inection (*n* = 44)	*p* Value
Age				
Range	1 M–99 Y	1 M–99 Y	6 M–89 Y	
Pediatrics (<15)	32 (16.67)	30	2	0.308
Aged (≥65)	79 (41.15)	62	17	**0.031**
Mean ± SD	54.61Y ± 28.14	52.94 ± 29.78	55.82 ± 22.14	
Gender: Male	117 (60.94)	91	26	0.398
Underlying Diseases and Comorbidities				
Malignancy	56 (29.17)	40	16	**0.041**
CNS and cerebrovascular diseases	36 (18.75)	31	5	**0.040**
Urinary tract infection	34 (17.71)	28	6	**0.044**
Hypertension	33 (17.19)	22	11	0.129
Cardiovascular diseases	28 (14.58)	24	4	**0.043**
Diabetes	25 (13.02)	20	5	0.281
Chronic kidney disease	20 (10.42)	20	3	0.391
Chronic pulmonary diseases	18 (9.38)	12	6	0.401
Chronic viral infections	11 (5.73)	8	3	0.677
Predisposing factors				
Invasive procedures	190 (98.96)	147	43	0.183
Intravenous catheter	116 (60.42)	79	37	**0.040**
Urinary catheter	46 (23.96)	41	5	0.129
Suction catheter	98 (51.04)	80	18	0.258
Endotracheal intubation	57 (29.69)	39	18	0.148
Gastrostomy (feeding) intubation	38 (19.79)	32	6	0.529
Surgery	56 (29.17)	40	16	**0.046**
Chemotherapy, Radiotherapy	19 (9.90)	12	7	**0.031**
Amputation	5 (2.60)	2	3	
Organ transplant	3 (1.56)	2	1	
Dialysis	5 (2.60)	5	0	
Hospitalization	190 (98.96)	147	43	
Medical wards				
General medicine	53 (27.60)	44	9	
Pediatric	33 (17.19)	31	2	
General surgery	24 (12.5)	14	10	
Emergency surgery	17 (8.85)	11	6	
Orthopedics	9 (4.69)	7	2	
Neurosurgery	3 (1.56)	3	0	
Others	51 (26.56)	37	14	
Patient in ICU of each ward	78 (40.63)	71	7	**0.001**

*p* value < 0.05 was considered to be significant; *p* value in bold letter = significant; M = Month; Y = Year; ICU = intensive care unit.

**Table 2 antibiotics-12-00410-t002:** Antibiotic susceptibility of *S. maltophilia* isolates.

Antibiotic	MIC (µg/mL)			Susceptibility (%)		
	MIC Range	MIC_50_	MIC_90_	S	I	R
Trimethoprim/ Sulfamethoxazole * (TMP/SMX)	0.047/0.893 → 32/608	0.19/3.61	0.5/9.5	185 (92.5)	0	15 (7.5)
Levofloxacin ** (LEV)	0.5 → 32	2	4	163 (81.5)	19 (9.5)	18 (9.0)
Ceftazidime ** (CAZ)	2 → 128	128	>128	21 (10.5)	24 (12.0)	150 (77.5)
Chloramphenicol ** (C)	4 → 128	16	32	67 (33.5)	97 (48.5)	36 (18)
Minocycline ** (MH)	0.5 → 4	0.5	2	200 (100)	0	0

MIC, A minimal inhibitory concentration value used breakpoint establishing by CLSI for *S. maltophilia*, document M100 ED29 [26]; MIC range, A minimal inhibitory concentration from the lowest value to highest value; MIC_50_, A minimum concentration value at which 50% of the isolates were inhibited; MIC_90_, A minimum concentration value at which 90% of the isolates were inhibited; * MIC determination was defined by MIC test strip; ** MIC determination was defined by agar dilution; S, Susceptible, I, Intermediate, R, Resistant; TMP/SMX, trimethoprim/sulfamethoxazole; LEV, levofloxacin; CAZ, ceftazidime; C, chloramphenicol; MH, minocycline.

**Table 3 antibiotics-12-00410-t003:** Antibiotic resistance genes among *S. maltophilia* isolates are stratified into six resistance groups.

Antibiotic Resistance	Antibiotic Resistance Genes								
	Total of Isolates *	*smeF*	*bla* _L1_	*bla* _L2_	*sul1*	*sul2*	*intI1*	*floR*	*mfsA*
		No. of Isolate (%)							
All isolates	200 (100)	183 (91.5)	86 (43)	20 (10)	12 (6)	4 (2)	14 (7)	8 (4)	9 (4.5)
TMP/SMX resistance	15 (7.5)	15 (100)	12 (80.0)	1 (6.67)	12 (80.0)	4 (26.67)	14 (93.33)	5 (33.33)	1 (6.67)
LEV resistance	18 (9)	16 (88.89)	7 (38.89)	1 (5.56)	4 (22.22)	2 (11.11)	5 (27.78)	2 (11.11)	2 (11.11)
CAZ resistance	157 (78.5)	123 (78.34)	80 (50.96)	19 (12.1)	12 (7.64)	4 (5.09)	13 (8.28)	12 (7.64)	7 (4.46)
C resistance	31 (15.5)	30 (96.77)	11 (35.48)	2 (6.45)	10 (32.26)	3 (9.68)	11 (35.48)	4 (12.90)	2 (6.45)
Non-resistance	37 (18.5)	37 (100)	6 (16.22)	1 (2.70)	0	0	0	5 (13.51)	0
MDR	20 (10)	20 (100)	11 (55)	1 (5)	10 (50)	3 (15)	12 (60)	4 (20)	2 (10)

Eight antibiotic resistance genes were detected among total *S. maltophilia* isolates and six resistant groups; * Number and percentage of the isolates of total and each antibiotic resistances; TMP/SMX, trimethoprim/sulfamethoxazole; LEV, levofloxacin; CAZ, ceftazidime; C, chloramphenicol; MDR, multidrug resistance.

**Table 4 antibiotics-12-00410-t004:** Virulence phenotypes of *S. maltophilia* isolates stratified on the different resistance groups.

Antibiotic Resistance	Biofilm Formation				Swimming Motility				Toxin and Enzymes			
	Non	Weak	Moderate	Strong	Non	Weak	Moderate	Strong	α-Hemolys in	β-Hemolys in	Protease	Lipase
	Number of Isolate (%)											
All isolates	1 (0.5)	22 (11)	31 (15.5)	146 (73)	21 (10.5)	73 (36.5)	81 (40.5)	25 (12.5)	106 (53)	94(47)	183 (91.5)	45 (22.5)
TMP/SMX resistance	0	2 (13.33)	6 (40)	7 (46.67)	6 (40)	8 (53.33)	0	1 (6.67)	11 (73.33)	4 (26.67)	11 (73.33)	2 (13.33)
LEV resistance	0	1 (5.56)	4 (22.22)	13 (72.22)	6 (33.33)	5 (27.78)	6 (33.33)	1 (5.55)	9 (50)	9 (50)	14 (77.78)	3 (16.67)
CAZ resistance	1 (0.64)	20 (12.74)	28 (17.83)	108 (68.79)	18 (11.46)	63 (40.13)	55 (35.03)	21 (13.38)	89 (56.69)	68 (43.31)	144 (91.72)	35 (22.29)
C resistance	0	5 (16.13)	7 (22.58)	19 (61.29)	8 (25.8)	12 (38.71)	9 (29.03)	2 (6.45)	15 (48.39)	16 (51.61)	25 (80.65)	8 (5.10)
Non-resistance	0	2 (5.4)	3 (8.10)	32 (86.49)	2 (5.4)	10 (27.02)	22 (59.46)	3 (8.11)	21 (56.76)	16 (43.24)	34 (91.89)	5 (13.51)
MDR	0	2 (10)	7 (35)	11 (55)	7 (35)	8 (40)	4 (20)	1 (5)	11 (55)	9 (45)	19 (95)	4 (20)

Three virulent phenotypes were detected among total *S. maltophilia* isolates and six resistant groups: TMP/SMX, trimethoprim/sulfamethoxazole; LEV, levofloxacin; CAZ, ceftazidime; C, chloramphenicol; and MDR, multidrug resistance.

**Table 5 antibiotics-12-00410-t005:** Genotypic and phenotypic characteristics of *S. maltophilia* isolates in each clonal complex.

Clonal * Complex	Sequence Types **	Specimens (Sources of Isolates)	Genotypes and Phenotypes Exhibited by Most of the Isolates				
			Drug ^a^ Resistance	Drug Resistance ^b^ Gene	Biofilm Formation	Swimming Motility	Toxin and ^c^ Enzymes Production
Group 1	3, 761, 762, 748, 750, 752, 753, 785, 796, 801, 807, 813, 814	Sputum, Fluid	1–2 drugs	1–2 genes	Strong producer	Weak swimming	2 types
Group 2	4, 27, 91, 363, 365, 613, 618, 619, 628, 634, 656, 660, 678, 686, 687, 697, 714, 737, 760, 770, 784	Sputum, Fluid, Pus, Urine	2–4 drugs	2–4 genes	Moderate and strong producer	Weak to moderate swimming	2 types
Group 3	28, 208, 212, 624, 626, 631, 647, 657, 658, 663, 665, 666, 668, 681, 700, 718, 720, 731, 738, 775, 791	Sputum, Fluid, Pus, Urine	1–3 drugs	0–5 genes (most isolates contained 2 genes)	Strong producer	Weak swimming	2 types
Group 4	621, 645, 648, 659, 667, 671, 673, 680, 685, 688, 697, 698, 705, 764, 789, 795, 802, 803, 808, 811, 812, 816, 818	Sputum, Fluid, Pus, Urine	0–1 drug	1 gene	Strong producer (All isolates)	Moderate swimming	2 types
Group 5	367, 605, 609, 627, 632, 643, 651, 692, 749, 754, 756, 757, 758, 759, 763, 766, 767, 773, 777, 781, 788, 790, 799, 810	Sputum, Fluid, Pus	1 drug	2 genes	Strong producer	Moderate to strong swimming	3 types
Group 6	376, 664, 669, 736, 745, 768, 769, 798, 805, 806, 817	Sputum, Fluid, Pus	1 drug	1 gene	Strong producer (All isolates)	Weak, moderate and strong (found in similar rate)	2–3 types

* Six clonal complex groups of *S. maltophilia* Thai isolates were classified using PHYLOViZ software; ** *S. maltophilia* STs belong to six clonal complex groups; ^a^ Number of drug resistance property; ^b^ Number of drug resistance genes; ^c^ Number of toxin and enzymes which were produced by each group.

## Data Availability

The new allele types and new sequence types of *S. maltophilia* in the present study were deposited in PubMLST: https://pubmlst.org/organisms/stenotrophomonas-maltophilia accessed on 7 December 2022.

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
