# Peer review of "Genotypic Diversity, Antibiotic Resistance, and Virulence Phenotypes of Stenotrophomonas maltophilia Clinical Isolates from a Thai University Hospital Setting"

_antibiotics, 2023, doi:10.3390/antibiotics12020410_

Round 1

Author Response

Dear Reviewer 1,

Thank you for the review and comments and for allowing us to improve our manuscript to Antibiotics. We appreciate the time and effort that you have dedicated to providing your valuable feedback on our manuscript. We have been able to incorporate changes to reflect most of the suggestions provided by the reviewers. We have highlighted the changes within the manuscript.

Yours sincerely,

Orathai Yinsai

Reviewer 2 Report

In the present study genotypic diversity, antibiotic susceptibility pattern and virulence phenotypes of 200 isolates  of  Stenotrophomonas maltophilia from a Thai University Hospital Setting has determined. The article is well written and the information is well presented. However, the studied isolates are from previous years (2015-2016) and the reported results are not up-to-date and may not be useful for controlling current isolates and clone. Therefore, it is necessary to study newer isolates and compare the results of their investigation with the existing results.  In addition, the demographic information of included patients and the results of their antibiotic therapy outcomes should be provided, and this information should be statistically studied with the results of bacteriological and genotyping tests. Authors should also specify how they distinguished clinical isolates from nonclinical isolates and contaminants.

Author Response

Dear Reviewer 2,

Thank you for the review and comments and also giving us the opportunity to improve our manuscript to Antibiotics. We appreciate the time and effort you have dedicated to providing valuable feedback on our manuscript. We have been able to incorporate changes to reflect most of the suggestions provided by the reviewers. We have highlighted the changes within the manuscript. 

Yours sincerely,

Orathai Yinsai

Round 2

Reviewer 1 Report

The Discussion section has not been SIGNIFICANTLY shortened. The Authors must take care of this before the manuscript can be considered newly for publication.

Author Response

Dear Reviewer 1,

We appreciate the time and effort that you have dedicated to providing your valuable feedback on our manuscript and giving us the chance to improve our manuscript again. We have been able to incorporate changes to reflect most of the suggestions provided by the reviewers. We highlighted the changes within the manuscript. Here are responses to the reviewers’ comments and concerns.

Comments and Suggestions for Authors:

The Discussion section has not been SIGNIFICANTLY shortened. The Authors must take care of this before the manuscript can be considered newly for publication.

Response: Thank you very much for reminding us about this point again. We have tried to shorten the Discussion section as you recommended and also added some points as suggested by reviewer 2. Changes of that section could be seen in our revised manuscript version 2.

This is all of the responses to your comments.

Please see the attachment to access full details of the changes. 

If you have another point to mention in this manuscript, please let us know

Yours sincerely,

Orathai Yinsai

Reviewer 2 Report

 The authors have not applied the suggestions and comments of the previous stage of the review. Results are not up-to-date and can not be useful for controlling current isolates and clone. The demographic information of included patients and the results of their antibiotic therapy outcomes have not been provided.

Author Response

Dear Reviewer 2,

Thank you very much for the comments and also giving us the opportunity to improve our manuscript again. We appreciate the time and effort you have dedicated to providing valuable feedback on our manuscript. We have been able to incorporate changes to reflect most of the suggestions provided by the reviewers. We have highlighted the changes within the manuscript. Here are responses to the reviewers’ comments and concerns.

Comments and Suggestions for Authors:

The authors have not applied the suggestions and comments of the previous stage of the review. Results are not up-to-date and can not be useful for controlling current isolates and clone.

Response: Thank you so much for suggesting this point again. Our results obtained from the highest prevalence period that may not up to date because the limitation of budget and human resources made us could not characterize more isolates in recent years. Additionally, in the latter years, S. maltophilia were found in a lower frequency that must be consume more time for collection and investigation. We also discussed this point in the discussion section.

The demographic information of included patients and the results of their antibiotic therapy outcomes have not been provided.

Response: We apologize that at first, we did not follow your recommendation due to the incomplete data. In this revised manuscript version 2, we tried to add the demographic information as you recommended as well as tried to shorten the discussion part as recommended by reviewer 1.  However, the therapy outcomes could not be included because the limitation of patient’s data accession.

This is all of the responses to your comments.

Please see the attachment to access full details of the changes. 

If you have another point to mention in this manuscript, please let us know

Yours sincerely,

Orathai Yinsai

Round 3

Reviewer 1 Report

The Authors satisfactorily replied to the comments I posed. Thanks.

Author Response

Dear Reviewer 1,

            We would like to express our appreciation for taking the time to give us your valuable suggestions which help us to improve our manuscript during the revision.  We are really glad to hear that our responses to your comments are being your satisfaction. We hope that this revised manuscript can now meet the reviewers' expectations and can be accepted for publication.

Thank you so much again

If you have another point to mention in this manuscript, please let us know

Yours sincerely,

Authors

Reviewer 2 Report

The authors have not applied the suggestions and comments of the previous stage of the review. 

Author Response

Dear Reviewer 2,

Thank you so much for your insightful comments on our study and providing such useful feedback. We appreciate for giving us the chance to improve our manuscript again. We have tried our best to change and explain our manuscript to satisfy your suggestion. We hope that this revised manuscript can now meet the reviewers' expectations and can be accepted for publication. Here is the response to your comment and concern.

Comments and Suggestions for Authors:

The authors have not applied the suggestions and comments of the previous stage of the review. Results are not up-to-date and can not be useful for controlling current isolates and clone.

Response: From your suggestions at the previous stage of the review, we really understand that using data from the recent isolates is necessary for infection control of current isolates and clone and it would have been interesting to explore this aspect. However, we believe that our data is still useful for infection control in terms of monitoring the global spreading of S. maltophilia using MLST based approach. Our study was initially aimed to characterize the isolates collected during the highest prevalence of S. maltophilia infection with the speculation of an outbreak within hospital. The results indicated that there was a high degree of genetic diversity among the isolates with no specific clone outbreak and that this species has a high genodiversity even when strains are isolated in the same hospital. We reported 131 novel ST types and deposited the new ST allelic profiles in open-access PubMLST (https://pubmlst.org/bigsdb?db=pubmlst_smaltophilia_isolates&page=query&prov_field1=f_country&prov_value1=Thailand&submit=1) for researchers who are interested in epidemiologic study of S. maltophilia and would like to compare their results with ours. To the best of our knowledge, our collection of ST allelic profiles is the first ever and only data from Thailand and none of those novel ST has been detected elsewhere indicating that there is still no global spreading of those STs.

Moreover, we would like to mention some of previous studies from China (collected isolates in 2014 and published in 2020) and Japan (collected isolates in 2005 and published in 2013) that also used not up-to-date isolates (references #32, #50) but their results are still useful for us to compare and monitor the spreading of S. maltophilia infection. In addition, providing the data from recent isolates in this manuscript may be not possible because, to do that, we have to apply for the new grant and S. maltophilia were found in a much lower frequency in the recent years. All those grant application processes would definitely consume more time for collection and investigation. We have indicated some limitations of our study in the discussion; line 382-387 Nevertheless, we also mentioned the recent study from Iran (references #55) which showed similar findings as our study in term of high genetic diversity of the isolates collection even when strains are isolated in one hospital (See the change at Discussion section; line 357 - 359). We assumed that whether we study the recent isolates, the result might not be significantly changed. S. maltophilia infection was not only transferred by patient to patient but also via hospital setting.  This fact leads to the idea for further study which will be investigated the genetic relationship between the isolates from patients and hospital equipment (This point also be discussed ;line 364 - 367).

This is all of the responses to your comments. We would like to inform you again that we really appreciate the time and effort that you and reviewer 1 have dedicated to correcting our manuscript. We have tried our best to apply all of the suggestions as much as possible and reviewer 1 is now satisfied with our revised manuscript. We look forward to hearing from you regarding our submission and to respond to any further questions and comments you may have.  

Yours sincerely,

Authors